# Inducible Nitric Oxide Synthase Embedded in Alginate/Polyethyleneimine Hydrogel as a New Platform to Explore NO-Driven Modulation of Biological Function

**DOI:** 10.3390/molecules28041612

**Published:** 2023-02-07

**Authors:** Shaimaa Maher, Lauren A. Smith, Celine A. El-Khoury, Haitham Kalil, Khalid Sossey-Alaoui, Mekki Bayachou

**Affiliations:** 1Chemistry Department, Cleveland State University, Cleveland, OH 44115, USA; 2Department of Medicine, Case Western Reserve University, Cleveland, OH 44106, USA; 3Metro Health Medical Center, Cleveland, OH 44109, USA; 4Department of Inflammation & Immunity, Lerner Research Institute, Cleveland Clinic, Cleveland, OH 44195, USA

**Keywords:** nitric oxide, release, hydrogel, biological function, nitric oxide synthase

## Abstract

Nitric oxide (NO), a small free radical molecule, turned out to be pervasive in biology and was shown to have a substantial influence on a range of biological activities, including cell growth and apoptosis. This molecule is involved in signaling and affects a number of physiologic functions. In recent decades, several processes related to cancer, such as angiogenesis, programmed cell death, infiltration, cell cycle progression, and metastasis, have been linked with nitric oxide. In addition, other parallel work showed that NO also has the potential to operate as an anti-cancer agent. As a result, it has gained attention in cancer-related therapeutics. The nitric oxide synthase enzyme family (NOS) is required for the biosynthesis of nitric oxide. It is becoming increasingly popular to develop NO-releasing materials as strong tumoricidal therapies that can deliver sustained high concentrations of nitric oxide to tumor sites. In this paper, we developed NO-releasing materials based on sodium alginate hydrogel. In this regard, alginate hydrogel discs were modified by adsorbing layers of polyethyleneimine and iNOS-oxygenase. These NO-releasing hydrogel discs were prepared using the layer-by-layer film building technique. The iNOS-oxygenase is adsorbed on the positively charged polyethyleneimine (PEI) matrix layer, which was formed on a negatively charged sodium alginate hydrogel. We show that nitric oxide is produced by enzymes contained within the hydrogel material when it is exposed to a solution containing all the components necessary for the NOS reaction. The electrostatic chemical adsorption of the layer-by-layer process was confirmed by FTIR measurements as well as scanning electron microscopy. We then tested the biocompatibility of the resulting modified sodium alginate hydrogel discs. We showed that this NOS-PEI-modified hydrogel is overall compatible with cell growth. We characterized the NOS/hydrogel films and examined their functional features in terms of NO release profiles. However, during the first 24 h of activity, these films show an increase in NO release flux, followed by a gradual drop and then a period of stable NO release. These findings show the inherent potential of using this system as a platform for NO-driven modulation of biological functions, including carcinogenesis.

## 1. Introduction

Since the identification of nitric oxide (NO) as the illusive endothelium-derived relaxing factor [1], it has been discovered that this simple molecule modulates a score of biological processes. In addition, NO is a hydrophobic free radical that diffuses easily through cell membranes [2]. The cardiovascular, neurological, and immunological systems are all regulated by nitric oxide. In the vasculature, platelet adhesion and aggregation, as well as smooth muscle cell (SMC) proliferation, are all inhibited by NO. Furthermore, NO plays a vital role in a variety of cardiovascular disorders, including atherosclerosis [3,4]. The endothelial, neuronal, and inducible nitric oxide synthase enzymes, abbreviated as eNOS, nNOS, and iNOS, respectively, all play a part in the biosynthesis of NO, and each of these enzymes is responsible for a different source site of NO. All known NOS enzymes function mainly in dimeric form when reducing equivalents from the reductase domain of one monomer activate the oxygenase domain of the other monomer. Arginine is the substrate of the NOS reaction, and its terminal guanidino nitrogen is oxidized by all three isoforms in a two-step reaction to produce NO and L-citrulline [5], Figure 1.

Numerous cell types, including macrophages and vascular smooth muscle cells (SMC), express the inducible NOS nearly exclusively in response to cytokines and lipopolysaccharides (LPS) [3,6,7].

Additionally, NO acts primarily through cGMP-dependent and cGMP-independent mechanisms. NO interacts with proteins that contain transition metals, such as heme-containing proteins. The iron in the heme group of soluble guanylyl-cyclase (sGC) is the main well-characterized target for nitric oxide. For a long time, NO was assumed to be carcinogenic since it has been shown to be involved in the activation of oncogenes, regulation of apoptosis and metastasis, DNA damage, inhibition of DNA repair, and tumor suppressor enzymes [6,7,8,9]. Nevertheless, there remains a point of contention when it comes to elucidating its modulation effects in cancer biology. This small molecule has the potential to either accelerate the development of cancer or inhibit its growth. In conditions where NO concentrations are lower (~100 nM), as in the case of endothelial NOS, NO is apparently associated with the maintenance of tumors because it promotes angiogenesis, which accelerates tumor propagation by providing needed blood supply and results in support for cell growth. However, higher amounts of NO (>400 nM), as shown in the case of iNOS, have the potential to promote phosphorylation of the p53 protein, leading to apoptosis in malignant cells. Moreover, NO can combine with superoxide to generate peroxynitrite. The latter is a very oxidative and nitrosative molecule that chemically modifies a number of signaling proteins directly [10,11]. The nitrosative signaling pathways are often activated in cancer cells to enhance growth and metastasis as well as treatment resistance, which often results in a poor prognosis [12]. On the other hand, peroxynitrite may trigger apoptosis and necrosis in cells through lipid peroxidation, cysteine oxidation, and protein nitrosylation [10,11].

The platforms that are biocompatible with cell culture and that can provide in situ defined fluxes of nitric oxide release on-demand have the potential to provide useful systems to study how nitric oxide, under various operating conditions, modulates processes and mechanisms of cell biology.

A number of polymer hydrogels, such as alginate hydrogels, have proven to be well suited for cell culture for various applications. The inspiration originally came from the components of the extracellular matrix, where cells from many tissues reside. The extracellular matrix is known to contain complex, three-dimensional materials that consist of elastic fibers made of various glycans and glycoproteins [13]. The alginate hydrogels have the advantage of undergoing gelation under gentle physiologic conditions, and their stable porous fiber network allows for easy diffusion of nutrients and other materials used by or released by attached cells.

Additionally, we previously introduced layer-by-layer (LbL) thin films of polyethyleneimine (PEI) and nitric oxide synthase (NOS) enzymes as composite coatings that release nitric oxide upon enzyme activation [14]. In this work, we combine the biocompatibility of alginate hydrogels with the LbL composite films of PEI/NOS to construct a hydrogel platform able to release nitric oxide on-demand under cell culture conditions. 

In this work, we use the oxygenase domain of inducible NOS (iNOSoxy) since this can be used to produce nitric oxide in one simple step using hydrogen peroxide as a source of reducing equivalents [14]. Our strategy is to electrostatically adsorb inducible nitric oxide synthase oxygenase on sodium alginate hydrogel and polyethyleneimine in a layer-by-layer (LbL) film building strategy. Additionally, iNOSoxy enzymes, as negatively charged species, are sandwiched between positively charged PEI layers coated on the alginate hydrogel. This iNOSoxy-based hydrogel construct has the potential to produce and release NO upon activation, which can be used to study its effects and modulation on cultured cells. For higher and prolonged NO flux release, the LbL technique could be employed to immobilize several NOS/PEI layers. In this study, we particularly focus on the examination and validation of the long-term stability of the NO-releasing hydrogel as a cell culture platform.

Alginates are naturally occurring, water-soluble linear polysaccharides extracted from several species of brown algae, such as Laminaria hyperborea and Macrocystis pyrifera, that are found worldwide in coastal waters [15]. Sodium alginate is one of the most widely used negatively charged polyelectrolytes that is rich in carboxylic groups due to its composition of β-D-mannuronic acid (M block) and α-L-guluronic acid (G block) units linked by β-1,4-glycosidic bond. In addition, alginate and its gel forms have a wide variety of applications. It has been extensively used as a biomaterial, which provides several advantageous features, including biocompatibility and non-immunogenicity. As stated earlier, alginate gels have been employed in a variety of applications, including wound healing, growth factor delivery, cartilage regeneration, cardiac remodeling, hepatic repair, and peripheral nerve regeneration [16]. Further, polyethyleneimine is a cationic synthetic polymer that contains several primary, secondary, and tertiary amine groups along its molecular chains [17,18]. The PEI itself is an excellent material for functionalizing and enhancing the properties of biomaterials in a variety of applications, including gene delivery [19], drug delivery [20], wound dressing [21], tissue engineering [22] and cell and enzyme immobilization [16,23]. 

The interaction of amine groups in the PEI structure with carboxylate groups in the alginate moieties has the potential to stabilize the composite platform (Figure 2). In this study, sodium alginate is used as an anionic base layer, while polyethyleneimine is used as the cationic polyelectrolyte layer. A layer of NOS enzymes (iNOSoxy in this case), which is negatively charged at neutral pH, is adsorbed on the external PEI layer to start the alternate PEI/iNOSoxy nitric oxide releasing layers.

## 2. Results and Discussion 

### 2.1. SA/PEI/iNOSoxy Hydrogel Preparation 

Alginate’s chemical structure and its ability to create gels have been extensively investigated [23,24,25]. When divalent cations such as Ca^2+^ are added to the guluronate blocks, the carboxylic groups engage in crosslinking that triggers the formation of a gel-like state (hydrogel). Alginates trap cations in a three-dimensional (3D) network that provides a stable, continuous, and thermo-irreversible 3D network. Properties such as swelling ratio and water content can be regulated independently by physical parameters such as cross-linking and polymeric chemical alterations [26,27]. 

After the formation of the alginate hydrogel, the crosslinking with a layer of polyethyleneimine follows, which will later allow the building of the PEI/iNOSoxy layers. We calculated the swelling ratio values of the SA and SA/PEI hydrogel discs using Equation (1). The findings indicate that SA and SA/PEI discs exhibit swelling properties above 92% (*w*/*w*) of water. This is not significantly different from the behavior of SA hydrogel. This suggests that PEI layers have no significant effect on the water content of the hydrogels [28].

### 2.2. FT-IR Measurements 

We used Fourier transform infrared spectroscopy (FTIR) to check the qualitative change in functional groups between the original sodium alginate and polyethyleneimine as controls versus the SA/PEI hydrogels (Figure 3). The FTIR spectrum of sodium alginate powder shows the expected presence of hydroxyl groups with a broad absorption band at 3326 cm^−1^. The absorption bands at 1404 and 1602 cm^−1^ are attributed to the symmetric and asymmetric stretching carbonyl groups of carboxylates (–COO–). The band at 2925 cm^−1^ is assigned to C–H stretching vibrations. The band at 1032 cm-1 is attributed to the stretching of the –C–O–C– groups [29]. The typical absorption peaks of amine groups-NH_2_ appear at 3270 cm^−1^ and 1627 cm^−1^ in the spectrum of authentic PEI. The stretching vibrations of the N–H and C–H groups are observed at 693 cm^−1^ and 2842 cm^−1^, respectively [27,30,31,32], Figure 3A. The FTIR spectrum of the sodium alginate hydrogel shows the typical bands of the –OH at 3363 cm^−1^ as well as the symmetrical and asymmetrical modes of the –COO groups at 1602 cm^−1^, and 1428 cm^−1^. Upon the incorporation of PEI onto the SA, the signature peaks of the sodium alginate hydrogel and polyethyleneimine layer were still observed. The peaks at 3270–3360 cm^−1^ corresponding to the O–H and N–H stretching vibrations, were amplified significantly and blended into a single broad peak, Figure 3B. This confirms the successful modification of the SA hydrogel disc with a PEI layer. The band positions in modified SAP discs shifted from 1602 to 1593 cm^−1^ and from 1428 to 1405 cm^−1^, respectively. The shift is attributed to hydrogen bonding between the N–H groups in PEI and the carboxylate –COO– groups in SA [33]. The FTIR characterization shows that the PEI layer is effectively attached to the SA, likely through simple electrostatic chemical adsorption.

### 2.3. Scanning Electron Microscopy/Energy-Dispersive X-ray Spectroscopic Characterization

After freeze-drying and gold-sputtering SA and SAP hydrogels, the surface morphologies were characterized by SEM. We also conducted EDX scans to determine the distribution of select elements in the hydrogel samples. SEM scans in Figure 4 show that the outer surface of SA is smooth and exhibits a slightly porous structure. This porous structure is expected and is the result of the three-dimensional network of alginate blocks that form coordination bonds with cations. Close analysis of the SA hydrogel (under magnification) shows continuous closed pores with relatively thick cell walls and a lamellar structure with ripples, as illustrated in Figure 4B,C. The structure is indicative of the underlying molecular networks of the hydrogel, which, upon drying and shrinking, result in the rippled features observed. The PEI-modified hydrogel discs show a clear morphological difference, as shown by a rough SAP surface with a higher porosity structure combined with a very thick cell wall and indentations (Figure 4D–F). The change in surface morphology of the SA and SAP hydrogels can be rationalized in terms of the expected chemical interaction and crosslinking of polyethyleneimine and sodium alginate [34]. The porous structure of SAP hydrogel will be advantageous for our NOS-modified hydrogels since it will allow relatively larger amounts of iNOSoxy to be adsorbed per surface area on PEI-modified hydrogel discs. 

EDX mapping analysis of the surfaces of SA and SAP hydrogels displays the content distribution of common elements such as carbon and oxygen. The surface of the SA disc contains C and O elements with no characteristic peak of nitrogen (Figure 5A). However, in SAP hydrogel discs, the peaks of C and N increased significantly compared to SA discs (Figure 5B). This finding suggests that the polyethyleneimine (PEI) layer has been successfully adsorbed on the surface of the SA hydrogel disc.

Furthermore, Figure 6 illustrates the difference in the nitrogen distribution peaks between the EDX spectra of SA and SAP. As expected, a new nitrogen peak arises in SAP because of the imine functionalities, but it is missing in SA.

### 2.4. Biocompatibility Testing of Alginate/PEI Hydrogel Using MDA-MB-231

Polymeric hydrogels, such as our SAP hydrogel, are made of water-soluble polymers. They are often biocompatible when they come into contact with blood, body fluids, and tissues. As a result, they are often employed as biomaterials in medical applications [35]. To evaluate the biocompatibility of the SAP hydrogel discs prior to adsorbing iNOSoxy, the SA and SAP hydrogel discs were co-cultured with the MDA-MB-231 cell line. A fast fluorometric approach was used to determine the viability of cells using acridine orange and propidium iodide dyes. The approach allows for the determination of viable and dead cells. When cells are visualized using fluorescence microscopy, viable cells fluoresce bright green, and nonviable cells fluoresce brilliant red. While the acridine orange and propidium iodide tests evaluate membrane integrity, the results are consistent with those obtained using other viability assays [36]. Acridine orange (AO) is a membrane-permeable, monovalent, cationic dye that binds to nucleic acids [37,38]. A low quantity of AO produces green fluorescence, whereas a large concentration produces red fluorescence. While propidium iodide (PI) is impermeable to intact plasma membranes, it rapidly penetrates dead cell plasma membranes [39] and intercalates with DNA or RNA to form brilliant red fluorescent complexes [32,40]. 

Upon staining with these dyes, images of hydrogels in cell cultures are acquired and compared. Figure 7 shows a comparative characterization of MDA-MB-231 cells cultured on sodium alginate (SA) and PEI-modified sodium alginate (SAP), as compared to controls and over different time periods (24 h to 72 h). Qualitatively, Figure 7 shows that cell viability is not significantly affected by sodium hydrogel (SA) or PEI-modified sodium hydrogel (SAP). In fact, one can visually see that cell viability on SAP is slightly higher than in the case of SA alone for all time windows.

A quantitative analysis of cell viability is provided in Figure 8. Cell counts are conducted and presented in the form of graph charts of percent viability. Again, the bar graphs show that cells on SAP hydrogel discs exhibit a higher survival rate compared to cells on SA hydrogels. However, both hydrogels do not show significant changes in cell viability compared to controls. Together, our results show that SAP discs are biocompatible with MDA-MB-231 cell growth.

### 2.5. Nitric Oxide Production Activity of SAP/iNOSoxy Hydrogel

Our SAP/iNOSoxy hydrogel discs successfully produced and released NO at rates greater than those reported for other inorganic NO-releasing materials in the literature [41]. We tracked NO fluxes in a cocktail buffer containing the substrate surrogate N-hydroxy-L-arginine in the presence of all other NOS reaction ingredients as described in the experimental section. The enzymatic reaction was conducted for the specified durations at 37 °C [42]. The SAP/iNOSoxy hydrogel’s enzymatic activity was assessed by measuring the quantity of NO released using the Griess assay, which monitors the buildup of nitrite, the breakdown product of NO. Due to the fact that the reaction cocktail contains all of the components necessary for the NOS reaction, the NO production and release are simply initiated by immersing the SAP/iNOSoxy hydrogel in the reaction cocktail. 

Figure 9A shows the cumulative catalytic production and release of NO into the reaction medium during the time of immersion. The initial increase in NO is rapid, but eventually reaches a plateau. For instance, the cumulative NO concentration released almost more than doubles during the first six-hour window (namely: 1.2 μM to 4.4 μM from 2 to 4 h, then grows to 9.4 μM in the 4 to 6 h-time range). The cumulative NO concentration continued to increase, but gradually, after 6 h, finally reaching a plateau after about 144 h. 

Another noteworthy functional characteristic of the nitric oxide releasing hydrogel is the flux of nitric oxide released with time. Figure 9B shows the computed average NO fluxes, i.e., the amount of NO released per unit surface area per unit time of our NOS-modified hydrogels, over the time course of the experiments. We used the NO concentration data points obtained at various times and the surface area of the NOS-modified hydrogels to calculate NOS fluxes. The graph shows an initial burst of NO production during the first 24 h, followed by a drop and then a consistent release of NO throughout the following 144 h. It is important to note that the reduced NO fluxes from our hydrogel are maintained and do not fall to zero, which implies that the activity of iNOSoxy enzymes embedded in the SAP hydrogels is maintained for longer periods of time past the 48 h. We previously addressed the observed reduction in NO flux [14] and concluded that no appreciable leaching of iNOSoxy into the buffer was observed under the conditions described. The drop in normalized NO production (flux) cannot be attributed to a leakage of iNOSoxy from the SAP hydrogel. Instead, a well-known long-term deactivation process, involving a NOS self-inhibition mechanism mediated by the enzyme’s product, NO, is likely the reason behind the drop in NO flux after the initial sharp burst in NO production [14]. Regardless, this multi-phasic NO release activity has no influence on the long-term NO release ability of our NOS-modified SAP hydrogels. 

In order to investigate the effect of the number of NOS/PEI layers on SAP hydrogel discs on observed NO fluxes, we analyzed NO release from SAP/iNOSoxy hydrogels with varying numbers of iNOSoxy layers. Figure 10 shows increased NO fluxes from a hydrogel with six layers of PEI/NOS as opposed to hydrogel discs with four layers of PEI/NOS. Data in Figure 10 shows that increased NO fluxes are achieved by increasing the embedded enzyme layers. Figure 10 also shows that the NO flux in a hydrogel with multiple PEI/iNOSoxy layers decreases relatively faster compared to hydrogels with one PEI/iNOSoxy layer. While the reason for this difference is not completely known, it is possible that it is due to a higher rate of iNOSoxy self-inhibition due to the relatively higher amounts of NO generated [14]. Regardless, the absolute numbers of NO flux in the hydrogels with multiple iNOSoxy layers are still larger than those of hydrogels with one PEI/iNOSoxy layer. The rate of NO release is crucial for the modulation of biological function. Hydrogels with a NO-releasing capability exceeding 400 µM at the surface of the hydrogel would be needed in our upcoming work to study NO-driven modulation of carcinogenesis. Therefore, it is vital to have the capability to control NO fluxes. In this regard, Figure 10 shows that we can adjust NO fluxes released from our SAP hydrogels by varying the number of NOS enzyme layers in the top polymeric PEI matrix. 

## 3. Materials and Methods

### 3.1. Materials 

An alginic acid sodium salt with low viscosity was obtained from MP Biomedicals. Branched polyethyleneimine (PEI, MW = 25 KDa) was obtained from Sigma-Aldrich (Saint Louis, MO, USA). Calcium chloride (CaCl_2_) was purchased from EMD Chemicals. Dr. Stuehr’s group at LRI of Cleveland Clinic generously provided the plasmid expressing the recombinant inducible nitric oxide synthase oxygenase domain (iNOSoxy) [43]. MDA-MB-231 cells were purchased from ATCC. Fetal bovine serum (FBS) and penicillin/streptomycin (P/S) were procured from Invitrogen, while DMEM was bought from Cytiva HyClone Laboratories. Acridine orange and propidium iodide stains were purchased from Aligned Genetics.

### 3.2. Preparation of Sodium Alginate/Polyethyleneimine/iNOS Hydrogel (Symbolized in This Work as SA/PEI/iNOSoxy)

#### 3.2.1. iNOS Oxygenase Domain Expression and Purification

The mouse iNOS oxygenase domain with a six-histidine tag was overexpressed in the ampicillin-resistant *E. coli* BL21 (DE3) strain using the pCWori vector plasmid and purified using Ni^2+^-nitrilotriacetate (NTA) affinity column chromatography as described before [14]. The concentration of protein was determined using Beer-Lambert law on an Agilent 8453 spectrophotometer by measuring the absorbance of the ferrous heme carbon monoxide adduct (extinction coefficient at 444 nm Ɛ_444_ = 76 mM^−1^ cm^−1^).

#### 3.2.2. Preparation of SA/PEI/iNOSoxy Hydrogel

Sodium alginate (2.0 g) was dissolved in 100 mL of deionized H_2_O to prepare a 2% (*w*/*v*) solution with continuous stirring until a complete dispersion of sodium alginate powder was achieved. Sodium alginate (SA) hydrogel discs with a diameter of 2–3 mm were made by punching small discs using disposable plastic core drills (~3 mm internal diameter) after immersing the homogenized hydrogel discs in a 5% (*w*/*v*) CaCl_2_ aqueous solution. The developed hydrogel discs are then immersed in CaCl_2_ with continuous stirring for 6 h to allow for complete cross-linking [34], followed by three washes with deionized H_2_O to remove excess CaCl_2_. The SA hydrogel discs were then placed in a solution of branched polyethyleneimine (PEI, 1.5 mg/mL) for 30 min with stirring. This process yields the hydrogel discs covered with a layer of PEI (symbolized as SA/PEI). After incubation, the hydrogel discs were thoroughly washed with deionized H_2_O. Finally, aliquots of 10 μL of the iNOSoxy solution as prepared were added on top of the SAP hydrogel for 15 min, followed by a final wash with deionized water, as shown in Figure 11.

#### 3.2.3. Water Content/Swelling Ratio of the SA/PEI Hydrogel as Prepared

Five SA and SA/PEI hydrogel discs were obtained as described above with diameters around ~3 mm. The SA and SA/PEI hydrogel discs were rinsed with deionized H_2_O three times and weighed as wet mass (M_wet_). Then, the discs were dehydrated using a vacuum and liquid nitrogen for 24 h. The dried hydrogel discs were then weighed again to determine the corresponding dry mass (M_Dry_). The percent swelling ratio was calculated using Equation (1) [28].
Swelling Ratio (%) = [(M_wet_ − M_Dry_)/M_wet_] × 100(1)

#### 3.2.4. FT-IR Characterization 

The infrared absorption spectra of sodium alginate, branched polyethyleneimine, as well as SA and SA/PEI hydrogels were obtained using the FT-IR spectrometer (PerkinElmer, L1050228, Waltham, MA, USA) within the scanning range of 4000–400 cm^−1^. The SA and SA/PEI hydrogels were collected after preparation, as shown above. The samples are washed three times with deionized water, frozen at −20 °C for 3 h, and then vacuum-dried with liquid nitrogen for an additional 24 h. The sample is then crushed and ground for FT-IR measurements.

#### 3.2.5. Scanning Electron Microscopy/Energy-Dispersive X-ray Spectroscopy

Freeze-dried SA and SA/PEI hydrogel discs were prepared as mentioned earlier, then mounted on aluminum stubs using electrically conductive carbon double-sided adhesive tape and vacuum coated with gold film using a sputter coater (SPI-MODULE, West Chester, PA, USA). A gold-coated hydrogel disc was examined for surface morphological characterization using a scanning electron microscope (INSPECT F50, FEI, Hillsboro, OR, USA). Additionally, the elemental composition of the SA and SA/PEI hydrogels was determined using energy-dispersive X-ray spectroscopy (EDX).

### 3.3. Biocompatibility Testing of Alginate/PEI Hydrogel Using MDA-MB-231

Prior to adsorbing the iNOSoxy layer on the SA/PEI hydrogel, the SA and SA/PEI hydrogel discs were tested for in vitro biocompatibility using the MDA-MB231 cell line. MDA-MB-231 is an epithelial cell line that was obtained from human breast cancer. A separate, smaller petri dish containing the same culture medium used for MDA-MB-231 was used to incubate the SA and SA/PEI hydrogel discs overnight. Cells were seeded in a 6-well plate at a density of 4 × 10^4^ cells per well and incubated under 5% CO_2_ in culture medium (90% Dulbecco’s Modified Eagle Medium). High glucose, 10% fetal bovine serum, 100 units/mL penicillin, and 0.01 mg/mL streptomycin solution at 37 °C until 60% confluence was reached. SA and SA/PEI hydrogel discs were applied for one day, two days, and three days. Cultured cells without hydrogel discs served as control groups to provide a baseline for comparison. Acridine orange (AO) and propidium iodide (PI) were used to stain the cells after each incubation period, which was followed by the replacement of the medium. Fluorescence on an inverted microscope was used to examine the labeled cells after 15 min of incubation in fresh medium. For each sample, at least five randomly selected images were used for counting to evaluate the MDA-MB-231 cell viability on SA and SA/PEI, as compared to the control [33]. ImageJ software was used to count the number of live and dead cells, and the viability of the cells was expressed as a percentage of the total number of cells after subtracting the number of dead cells.

### 3.4. Nitric Oxide Production Activity of SA/PEI/iNOSoxy Hydrogel

As described above, the 3-mm SA/PEI//iNOSoxy hydrogel discs were collected and incubated in a reaction buffer containing a source of reducing equivalents (H_2_O_2_ in our case) and the other usual NOS reaction ingredients (i.e., N^ω-^hydroxy-L-arginine and tetrahydrobiopterin) to monitor NO release and derive the corresponding NO flux. We used the Griess assay to determine the NO released [44]. The assay quantifies NO indirectly in the form of nitrite (NO_2_^−^) as a product of the autooxidation of NO [45]. The colored product of the Griess assay was monitored spectroscopically at 540 nm using the BioTek plate reader (BioTek 7201019, Winooski, VT). The experiment was performed as described previously [14]. Briefly, the assay was performed in triplicates in addition to a control sample using SA/PEI hydrogel without the iNOSoxy enzyme. Each hydrogel disc, including the control, was incubated separately in 1 mL of reaction buffer containing 1 mm N^ω-^hydroxy-L-arginine (NOHA), 100 μM tetrahydrobiopterin (H_4_B), and 0.5 mM dithiothreitol (DTT) in 100 mM phosphate buffer pH 7.4. The reaction was initiated by adding hydrogen peroxide to a final concentration of 30 mM at 37 °C. Aliquots of 50 μL of reaction buffer were collected from each vial and added to a 96-well microtiter plate at 2-, 4-, 6-, 24-, 48-, 72-, and 144-h lapse times. Aliquots were then analyzed using the Griess assay against a calibration curve constructed using aliquots with known concentrations.

## 4. Conclusions

We have successfully prepared and characterized new sodium hydrogel-based platforms (discs) with the capability to catalytically produce and release nitric oxide using embedded nitric oxide synthase (NOS) enzymes. In addition, we have shown that the hydrogel-based platform is biocompatible and does not significantly affect the growth of MDA-MB-231 cells. We further demonstrated that the NO-releasing capability of the NOS-modified hydrogel can be adjusted on-demand using varying numbers of NOS layers on the hydrogel discs. This will be very useful in our upcoming line of work focused on how varying amounts of NO released from NOS-based hydrogels affect biological functions in tissue cultures.

## Figures and Tables

**Figure 1 molecules-28-01612-f001:**
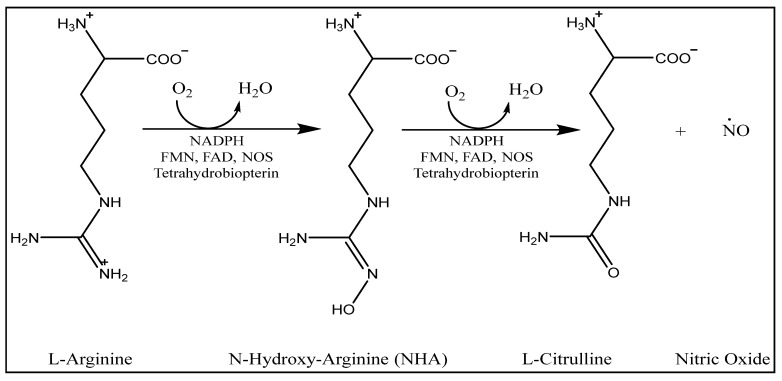
The general two-step catalytic process of NO biosynthesis by all isoforms of NOS enzymes. The first step converts L-arginine to N-hydorxy-arginine. The latter is converted to L-citrulline in the second step with the release of one equivalent of NO.

**Figure 2 molecules-28-01612-f002:**
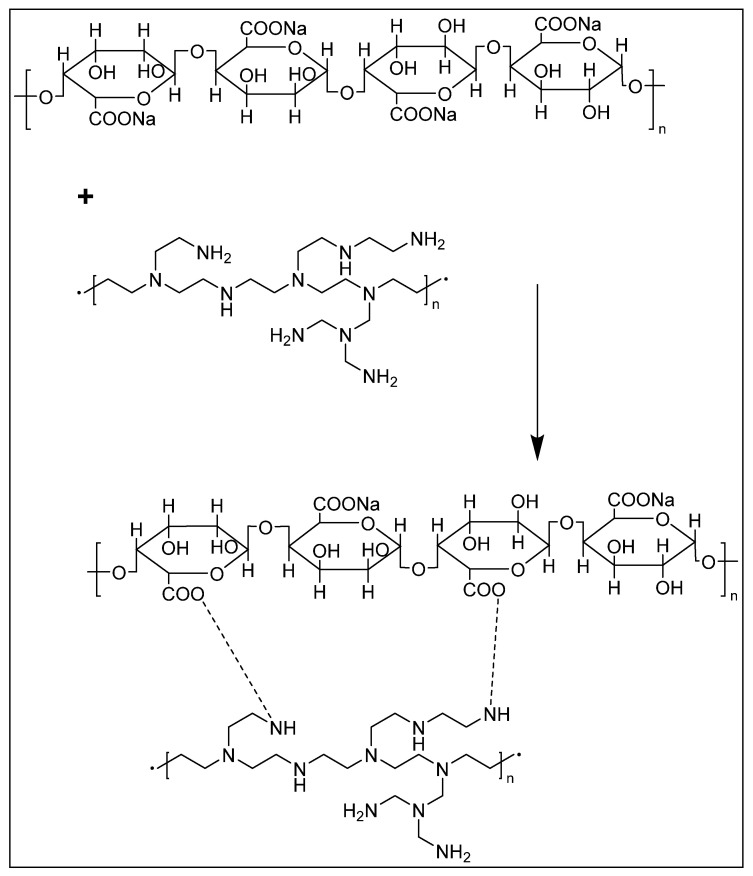
Schematic representation showing the cross-linking of sodium alginate and polyethyleneimine.

**Figure 3 molecules-28-01612-f003:**
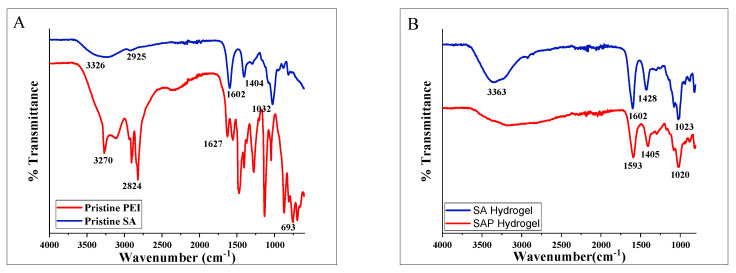
Fourier transform infrared (FTIR) spectra of (**A**) sodium alginate (blue trace) and polyethyleneimine (red trace), and (**B**) sodium alginate hydrogel (SA) and PEI modified sodium alginate hydrogel discs (SAP).

**Figure 4 molecules-28-01612-f004:**
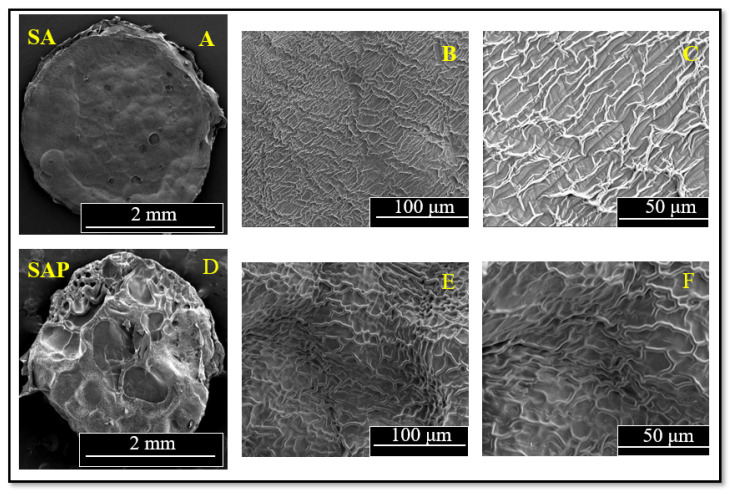
Scanning electron micrographs of sodium alginate hydrogel discs (SA) (micrographs (**A**–**C**), and of sodium alginate hydrogel discs modified with polyethyleneimine (SAP) (micrographs (**D**–**F**). Typical magnifications are: 50× for images (**A**,**D**), 1000× for images (**B**,**E**), and 2000× for images (**C**,**F**).

**Figure 5 molecules-28-01612-f005:**
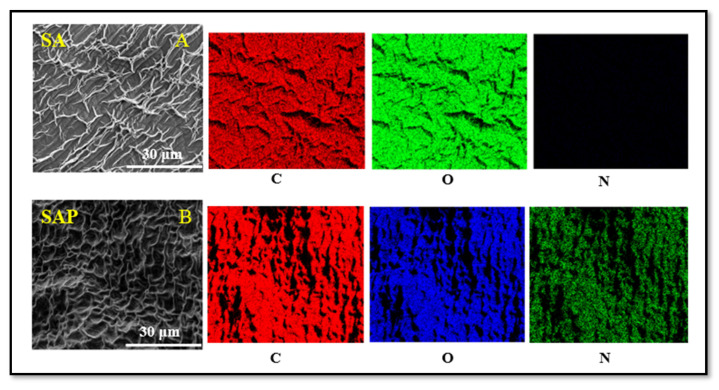
EDX mapping of carbon, oxygen, and nitrogen of sodium alginate hydrogel (SA) and of sodium alginate hydrogel modified with polyethyleneimine (SAP). The micrographs show the effective modification of the SA alginate hydrogel with PEI as shown with the significant increase of the nitrogen element in the EDX mapping.

**Figure 6 molecules-28-01612-f006:**
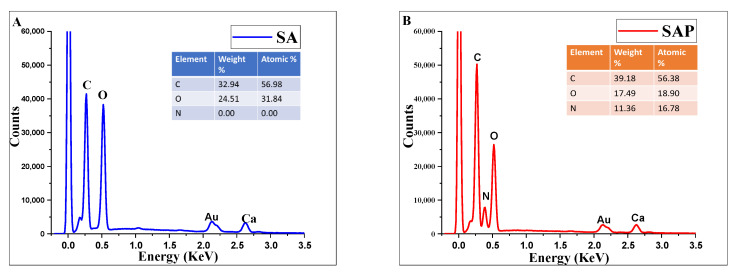
Surface Energy Dispersive X-ray (EDX) spectra of Sodium alginate and sodium alginate covered with PEI (**A**) EDX spectrum of SA and (**B**) EDX spectrum of SAP.

**Figure 7 molecules-28-01612-f007:**
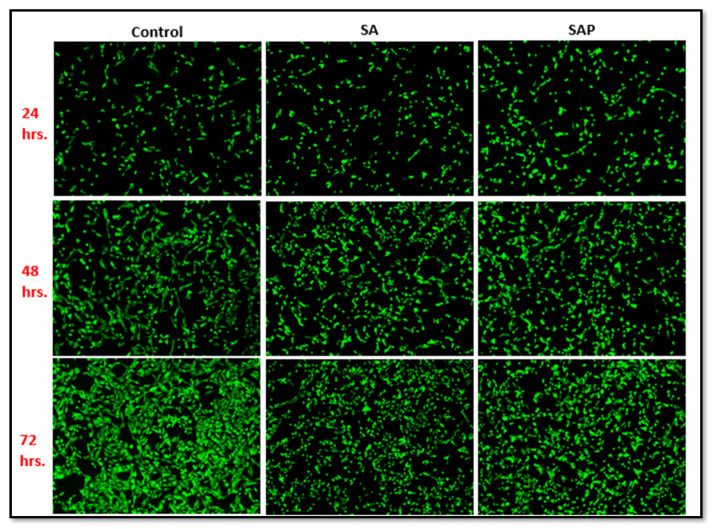
Fluorescence images of MDA-MB-231 cells stained with acridine orange (AO) and propidium iodide (PI) in the SA, SAP and control groups for 24 h, 48 h, and 72 h. The green spots staining with AO represent live cells, whereas the red dots staining with PI represent dead or dying cells.

**Figure 8 molecules-28-01612-f008:**
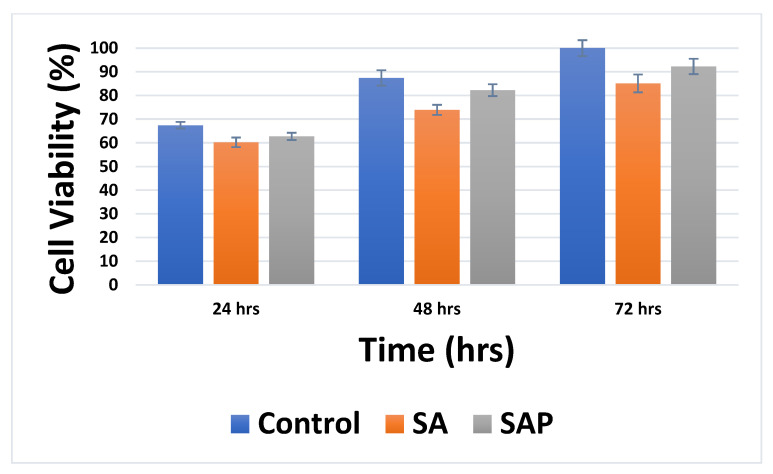
Survival rates of MDA-MB-231 cells on SA and SAP hydrogel discs. Bar charts represent mean percent viability in each case. Error bars represent ± standard deviation (SD for n = 3).

**Figure 9 molecules-28-01612-f009:**
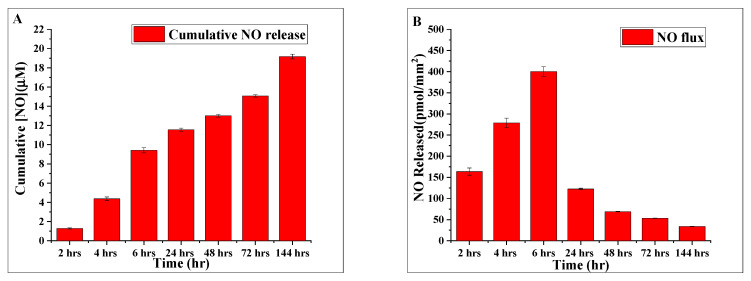
(**A**) Average cumulative NO concentration released from SAP/iNOSox hydrogel over time periods of experiments extending from 2 h to 144 h. (**B**) Average surface NO flux of SAP/iNOSoxy calculated in the form of amounts of NO released per time unit and per unit surface area. Data represents mean ± SD (n = 3).

**Figure 10 molecules-28-01612-f010:**
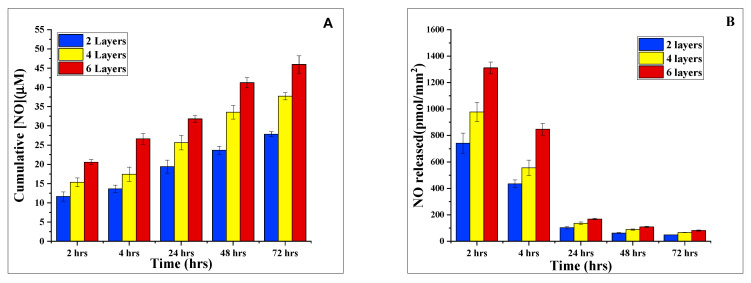
(**A**) Average cumulative NO concentration released from SAP/iNOSox hydrogel for various numbers of PEI/NOS layers over time periods of experiments extending from 2 h to 72 h. (**B**) Corresponding average surface NO flux of SAP/iNOSoxy for the various hydrogels calculated in the form of amounts of NO released per time unit and per unit surface area. Data represents mean ± SD (*n* = 3).

**Figure 11 molecules-28-01612-f011:**
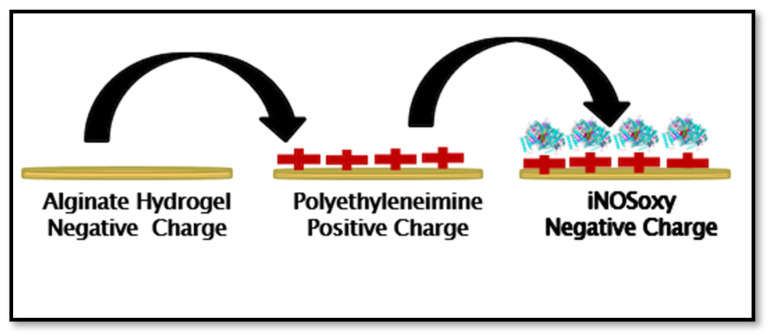
Preparation of SA/PEI//iNOSoxy hydrogel using crosslinking of the first layer of SA with PEI followed by additional layers of PEI/iNOSoxy.

## Data Availability

Data is contained within the article.

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
