# Peer review of "Inducible Nitric Oxide Synthase Embedded in Alginate/Polyethyleneimine Hydrogel as a New Platform to Explore NO-Driven Modulation of Biological Function"

_molecules, 2023, doi:10.3390/molecules28041612_

Round 1

Reviewer 1 Report

Inducible Nitric Oxide Synthase Embedded in Alginate/Polyethyleneimine Hydrogel as a New platform to Explore NO-driven Modulation of Biological Function

The nitric oxide-releasing hydrogel discs were prepared by layer-by-layer formation. The iNOS-oxygenase is adsorbed on the positively charged polyethyleneimine (PEI) matrix layer, which was bonded on top of a negatively charged sodium alginate hydrogel. The study shows that nitric oxide is produced by (i) enzymes (iNOSoxy) contained in the hydrogel material and (ii) exposed to a solution containing chemicals (H2O2 and Nω-hydroxy-L-arginine and tetrahydrobiopterin) for the NOS reaction.

Overall, this manuscript is a good report with sufficient information being provided.

Several technical issues may need the authors’ attention to further reinforce and enrich their report. 

1.      Authors should provide the magnification of all SEM images in Fig. 5. The ripple structures on the surface of hydrogel discs could be caused by the shrinking of materials as the liquid phase evaporates in the vacuum. The shrinkage can be estimated but it is somewhat related to the swelling ratio. Authors may be able to provide some qualitative explanations for this observation.

2.      SAP/iNOSoxy hydrogel discs released NO at rates that first increase and then reduce. However why the NOS/PEI multilayers on SAP hydrogel discs observed an almost exponentially decreased NO fluxes over time. I think there must be physical/chemical reasons that explain such a difference.

3.      Also, there is a little confusion from Figure 3 where the SA/PEI//iNOSoxy hydrogel uses crosslinking of the first layer of SA with PEI followed by additional layers of PEI/iNOSoxy. How about the multilayer configuration? Is it another PEI/iNOSoxy on top of the existing PEI/iNOSoxy? If so, what about the interface bond between PEI/iNOSoxy and PEI/iNOSoxy?

4.      In figure 8, ImageJ software was used to count the number of live (green) and dead cells (red), where are the red dots? What is the size of the area under optical images? The more rugged surface of SAP seems better suited for MDA-MB-231 cells. Any scientific reason for this observation?

Reviewer 2 Report

The paper deals with the new type of material which simulators biological function. The paper is clearly written and understandible and the results are relevant. 

I do not have any comments. Just the graphs from the release (Figures 10 and 11) copuld be the xy plot instead of bar plot in order to give to x-axis the real time scale.

Author Response

There was no specific request or question by this reviewer. We have included a sentence with the response to comments of Reviewer1